# Airport Group Operational Capacity Assessment and Analysis of Its Influencing Factors: Taking Typical Chinese Airport Group as an Example

Yafei Li and Meijun Teng *

College of Air Traffic Management, Civil Aviation University of China, Tianjin 300300, China
* Correspondence: 2021032056@cauc.edu.cn

**Abstract:** As an important transportation hub for air transportation, airports have played an important role in promoting regional economic and social development and improving the comprehensive national transportation system. The exploration of the key factors affecting the airport's operational capacity are of great importance to the sustainable development of the civil aviation transportation industry. In order to investigate the effect of airport operation, this paper selects 13 major airports in China's three major airport groups as the objects, defines the airport operational capacity by using entropy method combined with relevant indicators, calculates and sorts the operational capacity of sample airports, and analyzes the operational capacity of their internal airports by taking airport groups as units. By using the Tobit regression model, this paper analyzes the important factors that affect the operational capacity of sub-airports within the airport group. The results show that the economic level, urban development and the degree of opening to the outside world have a positive impact on the airport operational capacity. Different regional airport groups have different influencing factors on internal sub-airports' operational capacity.

**Keywords:** airport group; evaluation of airport operational capacity; entropy weight method; Tobit regression





## 1. Introduction

Looking at the experience of transportation planning at home and abroad, more and more single modes of transportation are changing to integrated transportation hub functions. Although aviation started late, it developed very rapidly. Because of its speed and maneuverability, it is an important way of modern passenger transportation, especially long-distance passenger transportation. As an important part of the air transportation system, airports play an important role in promoting regional economic and social development and improving the national comprehensive transportation system. Its planning should strive to form a perfect hub network, realize international and domestic large hub airport groups, and promote the scientific, healthy and balanced development of civil airports. In order to comprehensively and objectively evaluate the airport operational capacity, one must find out the key factors that affect its operational capacity, formulate relevant strategies to improve the airport operational capacity, improve relevant issues, and provide scientific decision support for the development of airport groups, which is of great significance to the sustainable development of civil aviation transportation industry.

Airport group is not a multi-airport system. The multi-airport system was innovatively put forward in 1970s–1980s, which refers to the airport collection consisting of two or more large airports and other airports in a metropolitan area [1]. The concept of "airport groups" was proposed by Chinese scholars in the 1980s and has not received much attention from the industry and academia since then. With the rapid development of urbanization, the study of airport groups as an important support for urban agglomeration, has attracted more and more attention [2]. With regard to the concept of airport groups, the National Civil Airport

Layout Plan promulgated and implemented in 2008 adopted the term "airport group" in the national documents for the first time and proposed to "build five regional airport groups with appropriate scale, reasonable structure and perfect functions". Feng [3], Director of Civil Aviation of China, pointed out at the 2017 China Civil Aviation Development Forum that the so-called "airport group" is not just a simple collection of airports in the region, but a multi-airport system characterized by coordinated operation and differentiated development. Airport group refers to a spatial cluster in a certain area, with one or more large airports as the core, which is formed by ground transportation links between airports and cities in the area based on aviation demand. It is worth pointing out that airport groups not only have the characteristics of a hierarchical structure, but also have the internal relationship of a network cascade.

In recent years, China's air transport market has grown rapidly and the airport scale has been continuously expanding; as of 2019, the number of transport airports in mainland China has reached 239, including 39 airports with tens of millions of passengers and nine airports ranking among the top 50 in the world (measured by passenger throughput). A number of characteristic airport groups are being formed around large hub airports. The three urban agglomerations of Beijing–Tianjin–Hebei, Yangtze River Delta and Pearl River Delta, with a land area of 3.6%, gather 18% of the national population and 35% of GDP. In 2021, the Beijing–Tianjin–Hebei airport group accommodated a total of 81.263 million passengers, the Yangtze River Delta airport group accommodated a total of 167.652 million passengers, the Guangdong–Hong Kong–Macao Greater Bay Area airport group accommodated a total of 87.241 million passengers and the Chengdu–Chongqing Airport group accommodated a total of 89.859 million passengers. The detailed data of the annual passenger throughput of the three airport groups from 2014 to 2020 are shown in Table 1.

**Table 1.** Annual passenger throughput of the three major airport groups in China (millions of people).

|  | 2014 | 2015 | 2016 | 2017 | 2018 | 2019 | 2020 |
|---|---|---|---|---|---|---|---|
| Beijing–Tianjin–Hebei Airport Group | 103.80 | 110.24 | 118.48 | 126.37 | 135.91 | 135.75 | 56.00 |
| Yangtze River Delta Airport Group | 147.98 | 164.78 | 181.16 | 198.50 | 214.49 | 225.15 | 133.33 |
| Pearl River Delta Airport Group | 95.13 | 99.63 | 107.84 | 120.63 | 130.29 | 138.59 | 89.01 |

So far, many scholars have studied and evaluated the airport service capability from the perspective of airport operation efficiency [4,5]. Scholars choose different input and output indicators and use the DEA model to calculate airport operation efficiency. In addition, the two-stage method has been used to evaluate airport performance in many studies. The so-called two-stage method, in the first stage, selects the airport's annual passenger, cargo throughput, take-off and landing flights, and other indicators when using the DEA model to evaluate the efficiency score of each airport; in the second stage, the Tobit regression model is used to measure the influence of multiple influencing factors on airport efficiency [6,7].

In this paper, we propose a method to rank the operational capacity of the sample airports. The entropy method is used to define the operational capacity of airports in combination with relevant indicators, the operational capacity of 13 sample airports is calculated and sorted and its changes with time from 2014 to 2020 are further analyzed. At the same time, in order to identify the factors that affect airport operational capacity, we initially selected eight explanatory variables and used the Tobit regression model for analysis. Taking into account the differences among airport groups in different regions, Tobit regression analysis is conducted again on the operational capability scores and variable indicators among sub-airports within the three airport groups, determining whether there are differences in the influencing factors of sub-airports' operational capacity in different regional airport groups.

This paper may be valuable to the government, civil aviation authorities and regulatory agencies. It is suggested that they consider the differences between the influencing factors of the internal airport operational capacity of different regional airport groups when making civil aviation strategic planning, so as to promote the coordinated development of local airport groups, surrounding cities, and other modes of transport.

The structure of the paper is as follows. In the second part, the literature review of airport efficiency evaluation and influencing factors analysis is introduced. Section 3 gives the research methods and models. The fourth part displays the indicators and data. Section 5 discusses the results obtained. Finally, Section 6 draws a conclusion.

## 2. Literature Review

As for airport performance evaluation, most scholars adopt nonparametric statistical methods. Data Envelopment Analysis (DEA) is a nonparametric statistical method to evaluate the "relative efficiency" of decision-making units proposed by A. Charnes in 1978 [8]. Until the end of the 1990s, some scholars used the DEA method to study airport efficiency. Lall first used the DEA model to study the operational efficiency of 21 airports in the United States from 1989 to 1993 [4]. Parker, Pels and others also used the DEA method to study the operational efficiency of airports in Britain and Europe, respectively [9,10]. Since then, more and more scholars have adopted the DEA model or the improved DEA model when performing airport efficiency evaluation. However, the choice of input and output items in the DEA method will have a decisive influence on the efficiency evaluation results. If the input and output items are not properly selected, the accuracy of the efficiency evaluation will be affected. The value of DEA evaluation efficiency is between 0 and 1, with 1 being effective and less than 1 being invalid. Additionally, there must be enough DMUs evaluated by the DEA method, otherwise most DMUs will be effective. In this paper, we want to study the degree of airport operational capacity rather than whether airport operation is effective, so we try to find another method besides DEA to define and calculate airport operational capacity

Entropy weight theory is an objective weighting method that is based on the concept of information entropy put forward by Shannon (1948). Using information entropy, the entropy weight of each index is calculated and then the entropy weight is modified according to each index, so as to obtain a more objective index weight. This method is commonly used in performance evaluation or risk evaluation of engineering technology and economic management. Liang et al. [11] designed a set of systematic evaluation methods to study the Chinese multi-level airport system, including the entropy method, an empirical weighting method. Shi et al. [12] aiming at the problems that statistical data causes for general aviation, unsafe events are limited and some indicators are difficult to quantify based on the AHP-entropy method, which weighted the subjective and objective combination and obtained the weights of each risk indicator, thus constructing the general aviation operation risk assessment system. Zhang et al. [13] established a grey calculation model of the airspace utilization rate of the terminal area based on the entropy method to quantitatively evaluate the airspace utilization rate of the terminal area. This model breaks the shortcoming of the traditional evaluation method, where the weight coefficient is not objective. Most scholars use the entropy weight method to build an evaluation index system. This paper attempts to use the entropy weight method combined with some indexes to define airport operational capacity.

In most of the literature, scholars try to find the key factors that affect airport efficiency after measuring airport performance. Usually, they adopt the two-stage method, that is, in the first stage DEA or other models are used to measure airport performance, and in the second stage, Tobit regression or other regression methods are used to analyze the factors affecting airport performance. Merkert et al. [14] used a truncated regression model in the second stage to evaluate the influence of ground competition on the efficiency scores of 35 Italian airports and 46 Norwegian airports in the first stage. The initial stage DEA results of Tsui et al. [15] show that Adelaide, Beijing, Brisbane, Hong Kong, Melbourne,

and Shenzhen are effective airports. The second stage regression analysis shows that the percentage of international passengers, the status of airlines, and the growth of GDP per capita are of great significance in explaining the change of airport efficiency. Zou et al. [16] investigated the impact of two main sources of funds used by American airports with airport production efficiency, airport improvement program (AIP) allocation and passenger facility cost (PFC). In the second stage, the random effect regression model was used, PFC and AIP were used as two explanatory variables, and the conclusion was drawn that PFC had a positive impact on airport efficiency, while AIP had a negative impact on airport efficiency. Ülkü [17] further determined the influence of management strategy and other external factors on airport efficiency according to the DEA efficiency scores of AENA in Spain and DHMI in Turkey obtained in the first stage, and finally, concluded that the airport network can be improved by closing some inefficient airports. Chaouk et al. [18] calculated the efficiency of 59 international airports in Europe and Asia-Pacific by the two-stage method and tested the influence of a group of macro-environmental factors on the efficiency.

The above studies all calculate and evaluate the efficiency of airports or airlines and its influencing factors but do not consider whether each airport belongs to different airport groups, the operating modes of different airport groups are slightly different, and whether there are differences in the factors that affect the operating capacity of their internal sub-airports. Thus, this paper uses the entropy method to define the airport operational capacity in combination with related indicators, analyzes the factors that affect the internal airport operational capacity of three major airport groups in China, and judges the differences of the internal influencing factors of different airport groups and compares their strengths.

## 3. Methodology

### 3.1. Using Entropy Weight Method (EWM) to Define Airport Operational Capacity Score

This paper defines the operational capacity of airports in three major Chinese airport groups using the entropy weight method, and sorts and compares the operational capacity scores of airports.

The theory of "entropy weight" is an objective weighting method. Shannon puts forward the concept of "information entropy" in 1948 [19]. Entropy is used to analyze the information provided by the data in the decision-making process [20]. When the values of various evaluation indexes are established after a given set of evaluation objects, entropy is the relative intensity of each index in the sense of competition. From the information point of view, it represents the amount of effective information provided by this evaluation index in this problem. As a comprehensive objective evaluation method, it mainly determines its weight according to the amount of information transmitted by each index to decision makers. In general, the smaller the information entropy of an index, the greater the variation degree of the index value, the more information it provides, the greater the role it can play in a comprehensive evaluation and the greater its weight. On the contrary, the greater the information entropy of an index, the smaller the variation of its value, the less information it provides, the smaller its role in a comprehensive evaluation and the smaller its weight.

(1) Standardize the data of each index

Assume that k indexes, $X_1, X_2, \ldots, X_k$ are given, where $X_i = \{x_1, x_2, \ldots, x_n\}$. $Y_{ij}$ is the value of the *j*th index of the *i*th sample after standardization.

The positive indicators are as follows:

$$Y_{ij} = \frac{X_{ij} - \min\{X_j\}}{\max\{X_j\} - \min\{X_j\}} \tag{1}$$

The negative indicators are as follows:

$$Y_{ij} = \frac{\max\{X_j\} - X_{ij}}{\max\{X_j\} - \min\{X_j\}} \tag{2}$$

(2) Solve the information entropy of each index

Based on the definition of information entropy in information theory, the information entropy of a group of data is $E_j = -\ln(n)^{-1} \sum\limits_{i=1}^{n} p_{ij} \ln p_{ij}$.

Where $p_{ij} = Y_{ij} / \sum\limits_{i=1}^{n} Y_{ij}$, if $p_{ij} = 0$, and then define $\lim\limits_{p_{ij} \to 0} p_{ij} \ln p_{ij} = 0$.

(3) The determination of the weight of each index

According to the calculation formula of information entropy, the information entropy of each index is calculated as $X_1, X_2, X_3, \ldots, X_n$. The calculation of the weight of each index by information entropy is as follows:

$$W_j = \frac{1 - E_j}{k - \sum E_j} (i = 1, 2, \ldots, k) \tag{3}$$

(4) To calculate the comprehensive score of airport operational capacity

$$S_i = \sum_{j=1}^{n} W_j \times p_{ij} \tag{4}$$

In the above formula $X_{ij}$ represents the value of the $j$ evaluation index of the $i$ airport, $\max\{X_j\}$ and $\min\{X_j\}$ are the maximum and minimum values of the $j$ evaluation index in all airports.

*3.2. Tobit Regression*

Tobit regression model was proposed by economist Tobin [21], also known as truncated regression model. Tobit regression is a model with limited explained variables, which is applicable in case the explained variable is the cut value or fragment value and is applied to the study of such limited data. The Tobit model, also known as sample selection model and constrained dependent variable model, is a model in which the value of the dependent variable meets certain constraints.

According to Tobin, the focus of restricted explained variables mainly consists of two aspects. One is the relationship between restricted dependent variables and other variables and the other is the hypothesis testing of this relationship. In the study of such problems, explanatory variables not only affect the probability of restricted explained variables but also affect the size of unrestricted explained variables.

When the entropy weight method is used to define and calculate the operation capacity of the airport, dimensionless processing (normalization) is required for the different index data units, so the result score is bounded at both ends of the 0–1 distribution. The data were truncated and the ordinary least squares method (OLS) was not suitable for estimating the regression coefficient. OLS was mainly used for parameter estimation of linear regression, and some values that minimized the sum of squares of the difference between the actual value and the model estimate were used as parameter estimators. Because OLS is used for linear regression of the whole sample, its nonlinear perturbation term is included in the perturbation term, resulting in inconsistent estimates. In contrast, the Tobit regression model, which based on the principle of maximum likelihood estimation, can process zero-value data and effectively avoid inconsistency and deviation in parameter estimation [22]. Therefore, the Tobit regression model of maximum likelihood estimation was used for regression analysis. The model is set as follows:

$$\begin{aligned} y_i^* &= x_i'\beta + u_i \\ u_i &\sim N(0, \sigma^2) \end{aligned} \tag{5}$$

$$y_i = \begin{cases} y_i^* & \text{if } y_i^* > 0 \\ 0 & \text{if } y_i^* \leq 0 \end{cases} \tag{6}$$

when the latent variable is $y_i^* \leq 0$, the explained variable $y$ is equal to 0; when $y_i^* > 0$, the explained variable y is equal to $y^*$ itself.

It is also assumed that the perturbation term $u_i$ obeys a normal distribution with a mean of 0 and a variance of $\sigma^2$.

In summary, research procedures and methods are shown in Figure 1:

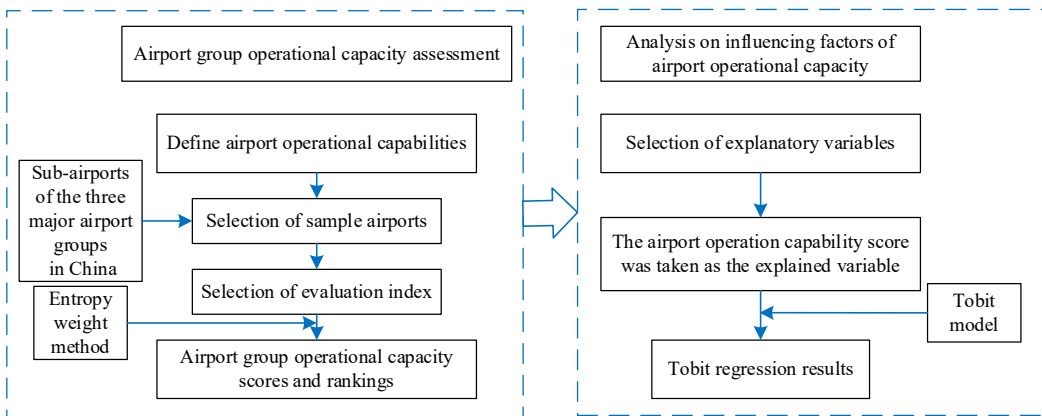

**Figure 1.** Research process and method.

## 4. Data

### *4.1. Indicators*

#### 4.1.1. Sample Airports

This study is based on panel data of thirteen major airports in the region from 2014 to 2020. Our sample consists of three major airport groups in China, including three major airports of the Beijing–Tianjin–Hebei airport group, seven major airports of the Yangtze River Delta airport group, and three major airports of the Pearl River Delta airport group. See Table 2 for details.

**Table 2.** Sample airports.

| Airport Group | Sub-Airport |
|---|---|
| Beijing–Tianjin–Hebei Airport Group | Beijing Capital International Airport(PEK) Tianjin Binhai International Airport(TSN) Shijiazhuang Zhengding International Airport(SJW) |
| Yangtze River Delta Airport Group | Shanghai Pudong International Airport(PVG) Shanghai Hongqiao International Airport(SHA) Hangzhou Xiaoshan International Airport(HGH) Nanjing Lukou International Airport (NKG) Ningbo Lishe International Airport (NGB) Hefei Xinqiao International Airport (HFE) Sunan Shuofang International Airport (WUX) |
| Pearl River Delta Airport Group | Guangzhou Baiyun International Airport(CAN) Shenzhen Baoan International Airport (SZX) Zhuhai Jinwan International Airport (ZUH) |

The geographical locations of the 13 airports are shown in Figure 2.

#### 4.1.2. Selection of Entropy Weight Index

In this paper, the definition of airport operational capacity should be based on the scientific principle, dynamics, and availability when selecting indicators. First of all, we should clarify the content and significance of various indicators and the selected indicators should reflect the connotation of such indicators scientifically and correctly. Generally, the data information will change with different periods, so we should pay attention to the

dynamic nature of the data. In the end, the selected index data should be available and the data sources should be accurate and reliable. Simultaneously, from as many different angles as possible, it reflects the airport operational capacity. This paper was focused on airport facilities, airport ground operation, and the number of routes opened. Selecting the terminal area, the number of parking spaces, the passenger throughput, the number of take-off and landing flights, the average daily flight volume and the number of routes, the above six indicators are combined with the entropy weight method to define the airport operational capacity and realize the measurement of the airport operation status. See Table 3 for descriptive statistics.

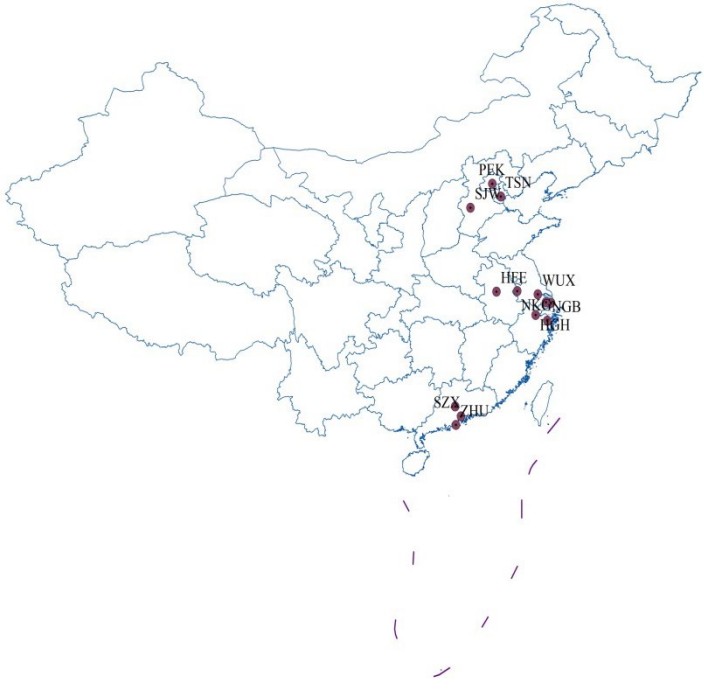

**Figure 2.** The geographical location of the sample airports.

**Table 3.** Descriptive statistical data of entropy weight index.

| | | Mean | Std.Dev | Min | Max |
|---|---|---|---|---|---|
| | Passenger throughput | $3.11 \times 10^7$ | $2.62 \times 10^7$ | 4,075,918 | $1.01 \times 10^8$ |
| Airport ground operation | Annual take-off and landing sorties | 227,364.4 | 169,947.7 | 35,781 | 614,022 |
| | Average daily flights | 1206.802 | 940.4802 | 138 | 3523 |
| Airport facilities | Terminal Area ($hm^2$) | 38.86436 | 48.46267 | 1.6567 | 141.4 |
| | Aircraft stands | 138.2308 | 108.2469 | 23 | 340 |
| Opened route | Routes | 183.5714 | 87.13975 | 42 | 400 |

### 4.2. Selection of Explanatory Variables in Tobit Regression Stage

As an important part of the transportation system, the airport's production and operation are influenced by many factors. In this paper, the airport operational capacity value calculated by the entropy weight method is used as the explained variable. In the last part, the six indexes selected when calculating the airport operational capacity by the entropy weight method will not be used as explanatory variables in the regression analysis of influencing factors, so as to avoid obtaining misleading or biased results [23]. We choose economic level, ground competition, urban development, foreign trade, and energy consumption as explanatory variables to construct the Tobit regression model, which follows the maximum likelihood estimation. The specific indicator names and descriptive statistics are shown in Table 4.

**Table 4.** Descriptive statistical data of Tobit regression influencing factors indicators.

| | Variables | Definition | Unit | Mean | Std.Dev. | Min | Max |
|---|---|---|---|---|---|---|---|
| Economic level | X1 | Per capita regional GDP | yuan | 113,489.5 | 34,842.43 | 39,919 | 203,489 |
| | X2 | Gross value of tertiary industry | 100 million yuan | 10,736.2 | 17,110.14 | 648.8385 | 183,339 |
| Ground competition | X3 | Passenger volume | 10,000 person-times | 26,434.26 | 31,467.78 | 1000 | 181,848 |
| | X4 | Goods volume | 10,000 tons | 32,586.64 | 18,762.7 | 4784.05 | 100,124 |
| Urban development | X5 | Permanent population | 10,000 person-times | 1213.387 | 692.8172 | 165.02 | 2487.09 |
| Foreign trade | X6 | Total export-import volume | 100 million dollars | 1841.314 | 1765.065 | 116.056 | 5374.74 |
| | X7 | Distance from airport to downtown | km | 24.68462 | 8.425584 | 10.6 | 35.8 |
| Energy consumption | X8 | Total Social electricity consumption | 100 million kwh | 683.5125 | 440.7512 | 100.4013 | 1575.96 |

The variables come from the statistical bulletin of national, economic, and social development from the official website, China Economic and Trade Network, and the cities where the airports are located.

The established Tobit regression model is:

$$Y_i = \beta_0 + \beta_1 X_1 + \beta_2 X_2 + \beta_3 X_3 + \beta_4 X_4 \\ + \beta_5 X_5 + \beta_6 X_6 + \beta_7 X_7 + \beta_8 X_8 + \varepsilon_i \tag{7}$$

Among them, $Y_i$ is the airport operational capacity of each airport from 2014 to 2020, $\beta_0$ is the intercept term, $\beta_1, \beta_2 \ldots \beta_8$ are the regression coefficients of explanatory variables, $X_i$ is the explanatory variable and $\varepsilon_i$ is the residual term.

## 5. Results

### 5.1. Efficiency Scores-EWM

In this section, the entropy weight method is used to calculate and sort the airport operational capacity of 13 selected sample airports, including the relative values of all sample airports from 2014 to 2020, and the relative operational capacity of internal sub-airports of the Beijing–Tianjin–Hebei airport group, the Yangtze River Delta airport group and the Pearl River Delta airport group is calculated and sorted, respectively.

5.1.1. Score of Relative Operational Capacity among 13 Sample Airports

In this section, the operational capacity of 13 sample airports is calculated first, and the results are shown in Table 5.

**Table 5.** Entropy weight method airport operational capacity score: single airport.

| | 2014 | 2015 | 2016 | 2017 | 2018 | 2019 | 2020 |
|---|---|---|---|---|---|---|---|
| PEK | 0.9583 | 0.9583 | 0.9570 | 0.9542 | 0.9535 | 0.9526 | 0.8319 |
| TSN | 0.1395 | 0.1476 | 0.1482 | 0.1695 | 0.1785 | 0.1687 | 0.1678 |
| SJW | 0.0656 | 0.0618 | 0.0618 | 0.0757 | 0.0803 | 0.0749 | 0.0797 |
| PVG | 0.6491 | 0.6923 | 0.6935 | 0.7318 | 0.7189 | 0.7390 | 0.7134 |
| SHA | 0.3289 | 0.3247 | 0.3001 | 0.3105 | 0.3091 | 0.3157 | 0.3926 |
| HGH | 0.2544 | 0.2668 | 0.2736 | 0.2879 | 0.2850 | 0.2951 | 0.3712 |
| NKG | 0.2487 | 0.2637 | 0.2803 | 0.2984 | 0.3014 | 0.3136 | 0.3547 |
| NGB | 0.0577 | 0.0558 | 0.0564 | 0.0628 | 0.0712 | 0.0720 | 0.0834 |
| HFE | 0.0394 | 0.0391 | 0.0357 | 0.0451 | 0.0483 | 0.0512 | 0.0567 |
| WUX | 0.0172 | 0.0169 | 0.0171 | 0.0182 | 0.0189 | 0.0189 | 0.0195 |
| CAN | 0.7860 | 0.7714 | 0.7841 | 0.8167 | 0.8244 | 0.8476 | 0.9571 |
| SZX | 0.4225 | 0.4350 | 0.4365 | 0.4383 | 0.4564 | 0.4744 | 0.6026 |
| ZUH | 0.0342 | 0.0322 | 0.0372 | 0.0494 | 0.0591 | 0.0624 | 0.0566 |

The resulting calculations using the entropy weight method, combined with six indicators, such as passenger throughput, take-off and landing sorties, and airport operational capacity, reflect the relative value of operational capacity among airports. The operational capacity values of each airport changing with the year are shown in Figure 3, which can clearly and intuitively show the operational capacity differences of each airport.

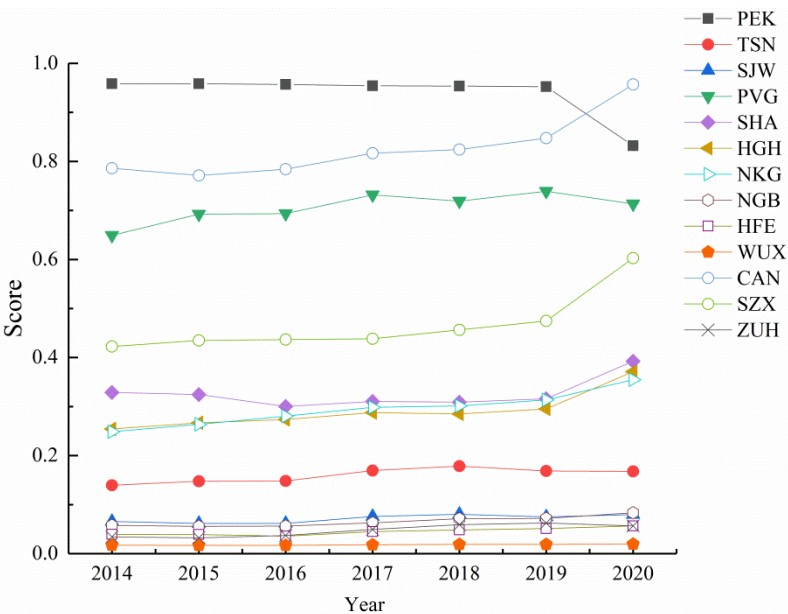

**Figure 3.** Time series changes of airport operational capacity of 13 sample airports.

It can be seen from the ranking results that from 2014 to 2020 the ranking of the operational capacity of 13 sample airports has not changed significantly and is relatively stable. These airports are PEK, CAN, and PVG. All the above airports belong to three Chinese gateway composite hub airports.

In 2014–2019, Beijing Capital International Airport's operational capacity scores were above 0.95 and ranked the highest, while in 2020, the score dropped to 0.832, ranking second only to CAN. We believe that in addition to the global COVID-19 outbreak, in 2020, the completion and start-up of the Beijing Daxing International Airport (PKX) at the end of 2019 also shared the passenger flow in Beijing, so that the operational capacity of PEK in 2020 was lower than before. The operational capacity of CAN ranks second among the 13 sample airports all year round, and its scores show a steadily increasing trend until 2020, when it receives the highest score of 0.957, surpassing PEK and ranking first; Shanghai is the first city in China to build an airport system of "one city and two airports" [24]. The term "one city and two airports" means that there are two civil airports in a city and the difference between the two airports in passenger transport and other indicators is not too great. This city is host to SHA and PVG, which have a clear division of labor. SHA in Shanghai is dominated by domestic routes, supplemented by international routes. While PVG covers almost all intercontinental routes from Shanghai, PVG also has some domestic flights, but it serves more business passengers [25]. In recent years, the rankings of PVG and SHA have remained unchanged, ranking third and fifth, respectively. In terms of airport operational capacity score, PVG has shown a relatively stable upward trend year by year, with a decline from 2019 to 2020. SHA also showed a relatively stable upward trend year by year, with the highest score of 0.393 in 2020. When the global epidemic broke out in 2020, the operational capacity of two airports in the same city changed differently, that is, the operational capacity of PVG changed from an increasing trend year by year to a decreasing one, while the operational capacity of SHA kept an increasing trend unchanged. As for this phenomenon, we think it has to do with the division of functions between the two airports. After the outbreak of COVID-19, the Chinese government issued a policy document on the

continuous reduction in intercontinental passenger flights during the epidemic prevention and control period. SHA is dominated by inland routes, supplemented by international routes, while PVG covers almost all intercontinental routes from Shanghai. Compared to SHA, the impact of the epidemic on PVG is greater. The operational capacity changes of SZX, HGH, and NKG are all in a steady and slight upward trend and all of them will reach their own maximum in 2020. This is not to say that the outbreak of the epidemic has played a positive role in improving the operational capacity of the above airports, but because we calculated the relative value of the operational capacity of each airport in that year, and the improvement of the scores was limited by the comparison among airports. The side shows that the impact of the COVID-19 epidemic on large hub international airports exceeds that of other large, medium, and small airports.

It should be noted that although the operational capacity scores of SJW, HFE, NGB, and WUX are close to 0, this does not mean that the operational capacity of the above airports is extremely low. As mentioned before, because we calculated relative scores of the operational capacity of 13 sample airports, our data results can reflect the relative size of the operational capacity of the above airports rather than the absolute value of the operational capacity of each airport.

5.1.2. Operational Capacity Scores among Sub-Airports within Each Airport Group

Through the calculation of airport operational capacity of three sub-airports in the Beijing–Tianjin–Hebei airport group, we obtained airport operational capacity scores of three airports from 2014 to 2020. The results are as shown in Figure 4 and Table 6.

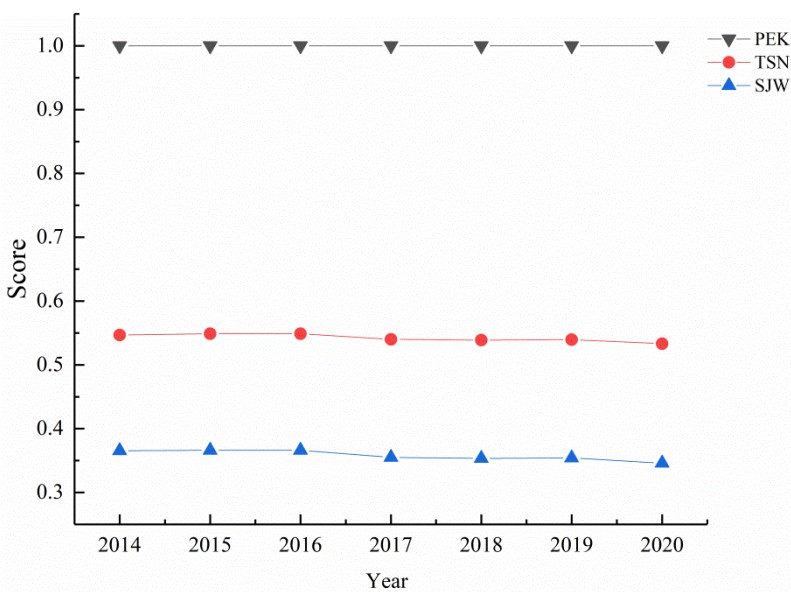

**Figure 4.** Time series changes of airport operational capacity of Beijing–Tianjin–Hebei airport group.

**Table 6.** Scores of operational capacity of airports within Beijing–Tianjin–Hebei airport group.

|  | 2014 | 2015 | 2016 | 2017 | 2018 | 2019 | 2020 |
|---|---|---|---|---|---|---|---|
| PEK | 1.0000 | 1.0000 | 1.0000 | 1.0000 | 1.0000 | 1.0000 | 1.0000 |
| TSN | 0.5468 | 0.5488 | 0.5488 | 0.5399 | 0.5389 | 0.5395 | 0.5332 |
| SJW | 0.3655 | 0.3662 | 0.3662 | 0.3551 | 0.3535 | 0.3542 | 0.3459 |

The operational capacity score of three airports in the Beijing–Tianjin–Hebei airport group are given in Table 6, respectively. From 2014 to 2020, the rankings of the three airports have not changed a bit. PEK ranks first in the Beijing–Tianjin–Hebei airport group and its score is still 1. TSN and SJW ranked second and third. Although the rankings did not

change, their scores changed, reaching their highest values in 2015 and 2016. Since 2017, their scores have gradually decreased, and the lowest were in 2020.

Results of airport operational capacity scores of seven sub-airports in the Yangtze River Delta airport group from 2014 to 2020 are shown in Table 7 and Figure 5, respectively:

**Table 7.** Scores of operational capacity of airports in Yangtze River Delta airport group.

|  | 2014 | 2015 | 2016 | 2017 | 2018 | 2019 | 2020 |
|---|---|---|---|---|---|---|---|
| PVG | 0.9930 | 0.9931 | 0.9931 | 0.9929 | 0.9927 | 0.9927 | 0.9922 |
| SHA | 0.5068 | 0.4638 | 0.4296 | 0.4209 | 0.4277 | 0.4219 | 0.5400 |
| HGH | 0.4000 | 0.3885 | 0.3988 | 0.3944 | 0.4003 | 0.3985 | 0.5136 |
| NKG | 0.3924 | 0.3945 | 0.4162 | 0.4198 | 0.4314 | 0.4370 | 0.5061 |
| NGB | 0.1035 | 0.0974 | 0.0967 | 0.1020 | 0.1158 | 0.1141 | 0.1338 |
| HFE | 0.0644 | 0.0612 | 0.0562 | 0.0660 | 0.0715 | 0.0734 | 0.0837 |
| WUX | 0.0271 | 0.0267 | 0.0266 | 0.0274 | 0.0285 | 0.0285 | 0.0302 |

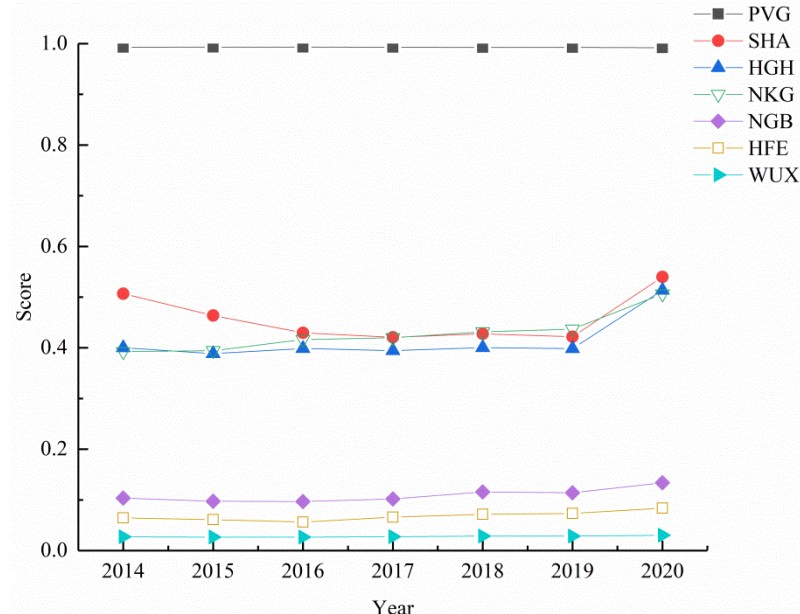

**Figure 5.** Time series changes of airport operational capacity of Yangtze River Delta airport group.

Based on the relative operational capacity scores and ranking results among the sub-airports in the airport group, the operational capacity of PVG has always been first among the seven airports, with a relatively stable score. There is no obvious change in the scores of NGB, HFE and WUX from 2014 to 2020. From 2014 to 2017, SHA ranked second in scores and was downgraded to third in 2018. Although the scores dropped, falling behind NKG at this time, there was a trend of slow increase by observing the running capacity scores of SHA. Since 2015, the ranking of HGH has dropped from the third place to the fourth place. The ranking remains until 2019, and then it returns to the third place in 2020. After the score of HGH decreased in 2015, there was a trend of slow recovery, but it still did not exceed NKG and SHA, which were, respectively located in the third place at that time. By 2020, it slightly increased to the highest score of the airport's operating ability in this period. The airport operational capacity score of NKG kept rising and it surpassed HGH in 2015 and rose to the third place, respectively, and surpassed SHA in 2018 and rose to the second place. From 2019 to 2020, NKG's ranking dropped to the fourth place. Careful observation of its airport operational capacity shows that although the score is increasing, it still does not exceed that of SHA and HGH.

Table 8 and Figure 6 respectively show the operational capacity scores and comparative rankings of the three airports in the Pearl River Delta airport group, which is similar to

those of the Beijing–Tianjin–Hebei airport group. From 2014 to 2020, the rankings of the three airports have not changed. CAN ranks first in the airport group in the Pearl River Delta, and its score is always 1. SZX and ZUH ranked second and third, both reaching their highest values in 2015. Since 2016, their scores have gradually decreased, with the lowest score in 2019. Compared with the previous year, there is an upward trend in 2020, which is different from the scores of the airports in the Beijing–Tianjin–Hebei airport group.

**Table 8.** Scores of operational capacity of airports in Pearl River Delta airport group.

|  | **2014** | **2015** | **2016** | **2017** | **2018** | **2019** | **2020** |
|---|---|---|---|---|---|---|---|
| CAN | 1.0000 | 1.0000 | 1.0000 | 1.0000 | 1.0000 | 1.0000 | 1.0000 |
| SZX | 0.8573 | 0.8597 | 0.8550 | 0.8498 | 0.8485 | 0.8477 | 0.8522 |
| ZUH | 0.5459 | 0.5463 | 0.5420 | 0.5358 | 0.5321 | 0.5306 | 0.5367 |

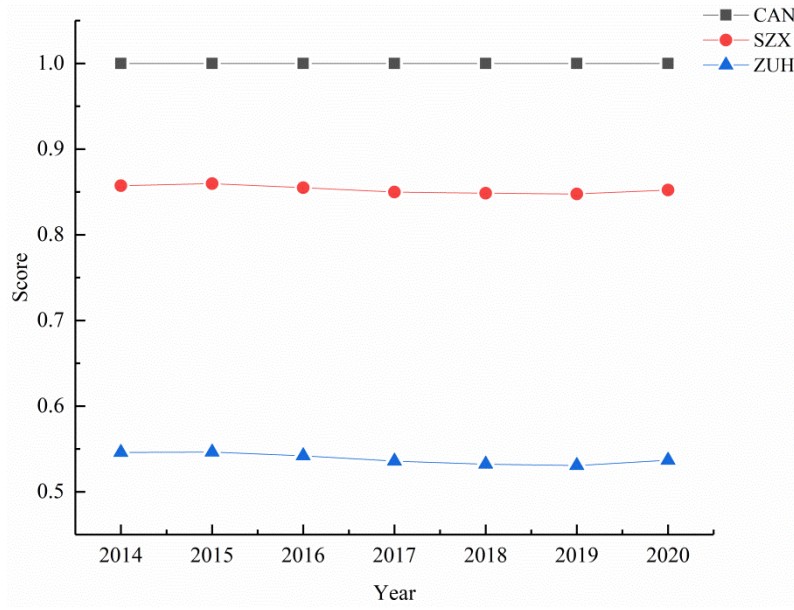

**Figure 6.** Time series changes of airport operational capacity of Pearl River Delta airport group.

There are three sub-airports in the Beijing–Tianjin–Hebei airport group and the Pearl River Delta airport group, and the ranking of their internal sub-airports' operational capacity has not changed and remained stable. However, there is a gap in the airports' operational capacity score. Beijing Capital Airport and CAN are in their own airport groups, and their airport operation ability scores are both 1 and ranked first. TSN ranks second in airport operational capacity among Beijing–Tianjin–Hebei airport groups, and its score fluctuates slightly around 0.54; SZX also ranks second among the airports in the Pearl River Delta, and its score fluctuates slightly around 0.85. This shows that although both of them rank the same in their own airport groups, the operational capacity of TSN is far behind that of PEK, and the operational capacity of SZX is closer to that of CAN, which ranks first in its airport group. Likewise, the third-ranked sub-airports in the two airport groups are SJW and ZUH, respectively. SJW's operational capacity score is kept at about 0.36, while ZUH's is heated at about 0.54. It is sufficient to show that the airport operational capacity in the Pearl River Delta airport group is smaller than that in the Beijing–Tianjin–Hebei airport group.

### 5.2. Tobit Regression Result

5.2.1. Regression Results of 13 Sample Airports

Table 9 shows the scores of 13 airports' operational capacity calculated by entropy weight method, which is the Tobit regression result as the explained variable.

**Table 9.** Tobit regression results of 13 sample airports.

| Explanatory Variable | Definition | Coef. | Std.Err | z | p |
|---|---|---|---|---|---|
| X1 | Per capita regional GDP | 0.0979231 | 0.0367547 | 2.66 | 0.008 |
| X2 | Gross value of tertiary industry | −0.0046357 | 0.0298425 | −0.16 | 0.877 |
| X3 | Passenger volume | −0.0408768 | 0.0244599 | −1.67 | 0.095 |
| X4 | Goods volume | 0.0179449 | 0.0547161 | 0.33 | 0.743 |
| X5 | Permanent population | 0.3070491 | 0.1568642 | 1.96 | 0.050 |
| X6 | Total export-import volume | 0.244186 | 0.0633752 | 3.85 | <0.0001 |
| X7 | Distance from airport to downtown | −0.2668644 | 0.1737639 | −1.54 | 0.125 |
| X8 | Total social electricity consumption | 0.0804093 | 0.0824647 | 0.98 | 0.330 |
| _cons | | 0.1849801 | 0.1206861 | 1.53 | 0.125 |

There are three variables that have passed the significance test: X1 (per capita GDP of the area where the airport is located), X5 (resident population of the city where the airport is located), and X6 (total import and export volume).

(1) The level of social and economic development is positively correlated with airport efficiency. Tobit regression results show that urban per capita GDP is under a significant positive effect on airport operational capacity, and its regression coefficient is 0.098. For every 1% increase in GDP per capita, the airport operational capacity can be correspondingly augmented by 0.098%. With the improvement of the economic level, the allocation of airport resources has been optimized, which can produce higher airport operational capacity.

(2) The variable of the number of permanent residents in the city is positively correlated with the airport operational capacity, and its regression coefficient is 0.307. The number of permanent residents in a city can reflect the scale and development degree of a city. If a city is relatively complete in scale and in a good state of progress, it will attract more residents to live. The increase in population will also increase the size of a city. Higher population density increases the demand for transportation, and the promotion of urban scale also means the improvement of resources in this area. Compared with other areas, its air transport industry and its supporting infrastructure will be more advanced. Air travel, as the quickest way to the outside world, can promote the exchange of a large number of people and goods between two larger cities in a short time, and the local airport operational capacity will improve.

(3) The degree of opening up to the outside world of the city where the airport is located can be reflected by the index of foreign trade. According to Tobit regression results, the influence coefficient of total import and export on airport efficiency is 0.244, which has a positive and significant impact on airport operational capacity. The improvement of the foreign trade level helps to increase the number of international routes and flights of local airports, thus improving the airport operational capacity.

In the previous section, the airport operation ability scores of the 13 sample airports and the selected eight explanatory variables were used for regression analysis. Considering the different development modes of urban agglomerations in different regions and the influence of airport groups on airports, we separately conduct Tobit regression analysis on the sub-airports in the three airport groups again. Taking an airport group as a unit, the operating capacity of each airport in the airport group obtained in Section 5.1 is the explained variable, and the same variable index as that in Section 5.2.1 is supposed to analyze the influencing factors.

### 5.2.2. Analysis of Regression Results of Sub-airports of Beijing–Tianjin–Hebei Airport Group

The Tobit regression results obtained by taking the sub-airports in the airport group as the research object are different from those of all the sample airports. Firstly, we studied three airports (PEK, TSN, and SJW) selected from the Beijing–Tianjin–Hebei airport group.

It can be seen from the regression results of the model in Table 10 that Tobit regression fitting is effective, which reflects the rationality for the selected variables to some extent.

**Table 10.** Tobit regression results of sub-airports of Beijing–Tianjin–Hebei airport group.

| Explanatory Variable | Coef. | Std.Err | z | p |
| :---: | :---: | :---: | :---: | :---: |
| X1 | −0.003147 | 0.0020107 | −1.58 | 0.114 |
| X2 | 0.0652705 | 0.0731586 | 0.89 | 0.372 |
| X3 | 0.0366571 | 0.009993 | 3.68 | <0.0001 |
| X4 | −0.0087173 | 0.0015012 | −5.81 | <0.0001 |
| X5 | 0.1897203 | 0.17276 | 10.98 | <0.0001 |
| X6 | 0.0172043 | 0.0124441 | 1.38 | 0.167 |
| X7 | 0.0201987 | 0.0095769 | 2.11 | 0.035 |
| X8 | 0.0353184 | 0.012432 | 2.84 | 0.004 |
| _cons | 0.6612115 | 0.0133666 | 49.47 | <0.0001 |

(1) Tobit regression results show that X3 (ground transportation of passengers) and X4 (ground transportation of goods), which are two indicators of ground competition, have a significant impact on the airport operational capacity in the Beijing–Tianjin–Hebei airport group, and both of them have passed the 1% significance level test. Their regression coefficients for airport operational capacity are one positive and one negative. The regression coefficient of X3 is 0.037, which indicates that there is a positive correlation between ground passenger transport and airport operational capacity, that is, for every 1% increase in ground passenger transport, airport operational capacity will increase by 0.037%; the regression coefficient of X4 is −0.0087, which indicates that there is a negative correlation between ground cargo transportation and airport operational capacity. For every 1% increase in ground cargo transportation, airport operational capacity decreases by 0.0087%, and there is a competitive relationship between ground cargo transportation and local air cargo transportation.

(2) The distance between airports in the Beijing–Tianjin–Hebei airport group is positively correlated with airport operational capacity, and Tobit regression coefficient of variable X7 airport distance to airport operational capacity is 0.020, which indicates that the greater the distance, the better the local airport operational capacity. The increase in the distance from the airport to the city center will theoretically increase the difficulty for local residents and surrounding passengers to arrive at the airport. At this time, taking ground transportation to the airport has become the choice of most passengers, and urban public transportation resources have been well utilized. However, the airport can avoid the peak of ground transportation if it is a long way from the city center.

(3) Energy consumption in the Beijing–Tianjin–Hebei region is positively correlated with the airport operational capacity and this influencing factor is reflected by the variable x8 electricity consumption of the whole society. The Tobit regression coefficient of total social electricity consumption to airport operational capacity is 0.035. The structure of power supply and demand in China has changed in recent years. From the demand side, the proportion of electricity consumed by residents and tertiary industries has increased, and energy consumption can drive economic growth. The more developed the economy, the more energy consumption. In combination with the influence of urban development, economic level and airport operational capacity, the airport operational capacity level increases by 0.035% for every 1% increase in energy consumption.

### 5.2.3. Regression Analysis of Sub-Airports in Yangtze River Delta Airport Group

The following conclusions can be drawn from the Tobit model regression results in Table 11: the urban development level in the Yangtze River Delta is positively correlated with the airport operational capacity of the city. Tobit regression results show that the regression coefficient of the number of urban permanent residents at the airport operational capacity in this area is 0.198, which has passed the significance level test of 1%. Every time

the resident population of a city in the Yangtze River Delta increases by 1%, the operational capacity of the local airport increases by 0.198%.

**Table 11.** Tobit regression results of sub-airports in Yangtze River Delta airport group.

| Explanatory Variable | Coef. | Std.Err | z | p |
|:---:|:---:|:---:|:---:|:---:|
| X1 | −0.0023625 | 0.0067386 | −0.35 | 0.726 |
| X2 | −0.0102552 | 0.184716 | −0.56 | 0.579 |
| X3 | −0.0041713 | 0.0061451 | −0.68 | 0.497 |
| X4 | 0.0455144 | 0.0116574 | 3.90 | <0.0001 |
| X5 | 0.1985291 | 0.0745311 | 2.66 | 0.008 |
| X6 | 0.0365519 | 0.0521169 | 0.70 | 0.483 |
| X7 | 0.1363571 | 0.0642579 | 2.12 | 0.034 |
| X8 | −0.412876 | 0.0286539 | −1.44 | 0.150 |
| _cons | 0.3562695 | 0.0626497 | 5.70 | <0.0001 |

### 5.2.4. Regression Analysis of Sub-Airports in Pearl River Delta Airport Group

When studying the three sub-airports in the Pearl River Delta airport group, we received the prompt that "independent variables are collinear with panel variables" on the stata14.0 software. After the collinearity test, three variables, X2 (gross value of tertiary industry), X6 (total import and export) and X7 (distance between airport and downtown), were eliminated. The results are as follows:

According to the Tobit model regression results in Table 12, we can obtain the following conclusions:

**Table 12.** Tobit regression results of sub-airports in Pearl River Delta airport group.

| Explanatory Variable | Coef. | Std.Err | z | p |
|:---:|:---:|:---:|:---:|:---:|
| X1 | −0.526846 | 0.0159601 | −3.30 | 0.001 |
| X3 | −0.0245765 | 0.010248 | −2.4 | 0.016 |
| X4 | −0.6033982 | 0.0418778 | −14.41 | <0.0001 |
| X5 | 0.17371 | 0.0380235 | 4.57 | <0.0001 |
| X6 | 0.1074513 | 0.0303234 | 3.54 | <0.0001 |
| _cons | 1.158337 | 0.0379499 | 30.52 | <0.0001 |

(1) The variable X3 ground transportation passenger transport (road passenger transport and railway passenger transport) and the variable X4 ground transportation cargo transport (road freight transport and railway freight transport) have a significant negative impact on the airport operational capacity in the Pearl River Delta airport group. This is because they both belong to the two indicators under the influence factor of ground competition, which indicates that land transportation and air transportation in the Pearl River Delta region have formed a competitive relationship. It is a double-edged sword to the operation of high-speed railway airport. With the distance, it can be divided into beneficial partners, complete competitors and non-influencers [26]. On the transportation market of 300–1000 km, high-speed rail is a complete competitor of civil aviation transportation. It takes part of the high-speed rail customers, which make the short-haul routes lose some passenger and cargo flows, resulting in the low running capacity of the local airport.

(2) According to the Tobit regression results, the influence factor of foreign trade is positively related to airport operational capacity. We select the variable of total import and export as the index to reflect foreign trade. The influence coefficient of this explanatory variable on airport operational capacity is 0.107, which passes the 1% significance level test.

It is worth noting that the variables X5 (resident population) have a good significance and all have a positive effect on the airport operational capacity when taking the airport group as a unit to explore the influencing factors of the sub-airports' operational capacity within the airport group. It is because the population is an important factor that determines

the amount of urban residents' travel. The Tobit regression coefficient of resident population to airport operational capacity is 0.1897 in the Beijing–Tianjin–Hebei airport group, 0.1985 in the Yangtze River Delta and 0.1737 in Pearl River Delta. The increasing degree of the resident population of the city to the airport operation ability score is Yangtze River Delta > Beijing–Tianjin–Hebei > Pearl River Delta. It can be observed that residents in the Yangtze River Delta cities have a stronger willingness to travel by air than those in the Beijing–Tianjin–Hebei region and the Pearl River Delta region, followed by the Beijing–Tianjin–Hebei region, and the Pearl River Delta region is the lowest. With the increase in the number of permanent residents, the number of people who choose to travel by air in the Yangtze River Delta is higher than that in the other two regions, and the operational capacity of sub-airports in the Yangtze River Delta airport group is higher than that in the Beijing–Tianjin–Hebei airport group and the Pearl River Delta airport group. It is estimated that the developed road network and high-speed rail network in the Pearl River Delta make the local population choose road and rail transportation more.

*5.3. The Coupling Analysis between the Operation Capacity of Regional Airports and the Economic and Social Development Level of the Region*

With reference to the existing research results [27], the coupling analysis between the operation capacity of sample airports and the regional economic and social development level was carried out. Index selection is shown in Table 13.

**Table 13.** Evaluation index system of the operation capacity of regional airports and the economic and social development level of the region.

| Destination Layer | Schematic Layer |
|---|---|
| Airport operational capacity (A) | Passenger throughput (a1) Cargo throughput (a2) Aircraft take-off and landing (a3) |
| The level of economic and social development (B) | Per capita GDP (b1) Total export-import volume (b2) |

The value of coupling coordination is between 0 and 1. The closer the value is to 0, it indicates that the worse the coupling coordination degree of subsystems is worse, and the benign interaction ability between airport operation capacity and regional economic and social development level is weaker. The closer the value is to 1, the better the coupling coordination degree of each subsystem, and the stronger the benign interaction ability between airport operation capacity and regional economic and social development level. The coupling coordination degree can be divided into five levels, as shown in Table 14.

**Table 14.** Coupling coordination degree classification.

| Coupling Coordination Degree | Coordination Type |
|---|---|
| 0.0–0.2 | Severe disorder |
| 0.2–0.4 | Moderate disorder |
| 0.4–0.6 | Reluctant coordination |
| 0.6–0.8 | Moderate coordination |
| 0.8–1.0 | Good coordination |

According to the calculation results, the characteristics of the coupling and coordinated development between airport operation capacity and regional economy and society are further analyzed. As can be seen from Figure 7, from 2014 to 2020, the coupling coordination degree of most airports shows a general downward trend. Except for WUX, the operating capacity of other airports and the coupling coordination degree of the regional economy and society all reach the lowest value in 2020.

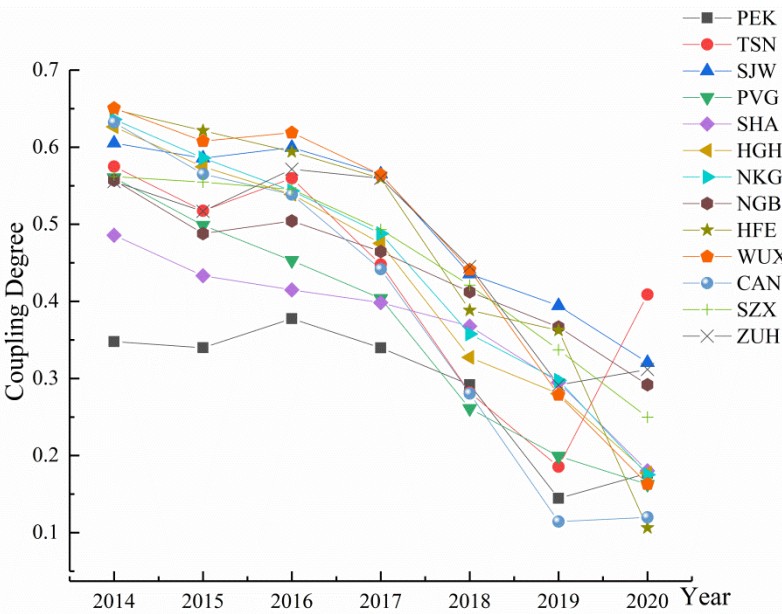

**Figure 7.** The operational capacity of 13 sample airports and the coupling coordination degree of regional economy and society.

## 6. Conclusions

In this paper, we used the entropy method to define airport operational capacity and explore the key factors that affect airport operational capacity. This paper selects six indexes, including passenger throughput, number of take-off and landing flights, terminal area, parking spaces, number of routes and daily average flight volume, and defines an evaluation index-airport operation capacity, which can reflect the development and operation effect of the airport by combining the entropy weight method. Not only is the operation capacity of 13 selected sample airports calculated and sorted, but also the operation capacity of their internal airports is calculated and sorted by taking airport groups as units, and the analysis is made in combination with the present situation.

In order to further explore the factors that affect the airport's operational capacity, we preliminarily judge that economic level, ground competition, urban development, foreign trade and energy consumption can affect the airports' operational capacity and the selected eight indicators. By using the Tobit regression model, we analyze the influencing factors of the relative operational capacity among all 13 sample airports. Among the eight explanatory variables we selected, three variables have strong significance. This result gives us the reasons that affect the airport's operational capacity: GDP per capita has a significant positive effect on the airport's operational capacity; improving the economic level and optimizing the allocation of airport resources can produce better airport operational capacity; both the development of the city and the degree of opening to the outside world have a positive impact on the operational capacity of the local airport. Envelope et al. [28] conducted a study on the efficiency of some airports in Spain and believes that tourism is an important influencing factor, and puts forward suggestions to increase the routes connecting regional airports with major countries and international tourism markets, so as to improve airport efficiency with the development of tourism as the orientation. Kaya et al. [29] has found that the efficiency of Turkish airports varies by region. In fact, airports based in the Mediterranean region are significantly more efficient than those in other regions. An increase in the number of tourists and museum sites in the city can make airports more efficient. Yilmaz et al. [30] found through the analysis results of the Tobit regression model that two significant factors, the number of tourists and the distance to the city center, can explain the changes in the efficiency of Turkish airports. Airports located in high-density tourist areas are expected to obtain higher efficiency scores, and airports reasonably close to the city center make transportation more convenient. Then, we consider

whether there will be differences in the factors that affect the operational capacity of its sub-airports for different airport groups in different areas. Therefore, we carried out Tobit regression analysis on the operational capacity scores among the sub-airports in the three airport groups and the eight explanatory variables initially selected. One of the important indicators of a world-class urban agglomeration is that it has many international external transportation hubs and distinct hub systems. Correspondingly, for the airport group, it is necessary to strengthen the function of its core airport as a large-scale international aviation hub and form a relatively balanced regional aviation hub layout with other airports in the cluster. Based on the experience of the coordinated development of world-class airport groups, such as New York and London, the core of the coordinated operation of airport groups lies in the clear positioning of airports and differentiated development. It is a key measure to improve the operation capability of airport groups to realize the dislocation development of airport groups.

The limitation of this study is that the selection of airport operational capacity evaluation index may not be comprehensive enough to reflect all aspects of airports' capabilities or levels, and the evaluation index system needs to be further improved.

In this paper, the index data of 13 sample airports in three major Chinese airport groups from 2014 to 2020 were selected for calculation. Beijing Daxing International Airport (PKX) was completed and put into use in 2019. This paper does not include PKX in the study. The completion and operation of PKX mean that a new pattern of "one city, two games" has been formed in the capital Beijing, and the coordinated development of Beijing–Tianjin–Hebei airport group and Beijing–Tianjin–Hebei urban agglomeration will be affected to some extent. In the future, new index data can be obtained to study this.

**Author Contributions:** Conceptualization, Y.L. and M.T.; methodology, Y.L. and M.T.; software, M.T.; validation, M.T.; formal analysis, M.T.; writing—original draft preparation, M.T.; writing—review and editing, Y.L.; visualization, M.T.; supervision, Y.L.; funding acquisition, Y.L. All authors have read and agreed to the published version of the manuscript.

**Funding:** This work was partially supported by MOE (Ministry of Education in China) Youth Fund Project of Humanities and Social Sciences (Grant No. 21YJCZH075); General Program of Tianjin Applied Basic Research Diversified Investment Fund (Grant No. 21JCYBJC00720).

**Institutional Review Board Statement:** Not applicable.

**Informed Consent Statement:** Not applicable.

**Data Availability Statement:** Not applicable.

**Conflicts of Interest:** The authors declare no conflict of interest.

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
