# Peer review of "Airport Group Operational Capacity Assessment and Analysis of Its Influencing Factors: Taking Typical Chinese Airport Group as an Example"

_sustainability, doi:10.3390/su15021654_

Round 1

Reviewer 1 Report

In this paper, the authors analyze the operational capacity of airports by using the entropy method with relevant indicators and a Tobit regression model. Overall, the article seemed interesting and I have the following comments:

- Although the paper relies heavily on the Tobit model, little to no details are provided on this model. It would help to provide some background on this model.

- The likelihood formulation of Tobit as compared to an Ordinary Least Squares regression can be better explained and the advantages of the Tobit model can also be compared while discussing the results.

- There are some sentences in the paper with language errors. Please rectify these.

Author Response

Reviewer 1

In this paper, the authors analyze the operational capacity of airports by using the entropy method with relevant indicators and a Tobit regression model. Overall, the article seemed interesting and I have the following comments:

- Although the paper relies heavily on the Tobit model, little to no details are provided on this model. It would help to provide some background on this model.

- The likelihood formulation of Tobit as compared to an Ordinary Least Squares regression can be better explained and the advantages of the Tobit model can also be compared while discussing the results.

- There are some sentences in the paper with language errors. Please rectify these.

Response:

Thanks very much for the comments, which are very helpful to improve the quality of this article. We have revised the manuscript and especially paid much attention to your comments and suggestions.

  1. We have added the background knowledge of the Tobit model, highlighted the reasons for choosing the Tobit model and added the details of the model. (The specific modification is located in section 3.2 on page 5)
  2. We have compared OLS and Tobit model and expounded the advantages of choosing Tobit. (The specific modification is located in section 3.2 on page 5)
  3. We have corrected the grammatical mistakes in the text as much as possible.

Reviewer 2 Report

1. The font type of the chart axis should be unified with Figure 1 and other charts, and Equation 6 is not formatted correctly.

2.It is recommended to supplement the coupling analysis between the operation capacity of regional airports and the economic and social development level of the region.

3. Some statements in the text need to be modified, and there are some errors in the syntax.

Author Response

Reviewer 2

  1. The font type of the chart axis should be unified with Figure 1 and other charts, and Equation 6 is not formatted correctly.
  2. It is recommended to supplement the coupling analysis between the operation capacity of regional airports and the economic and social development level of the region.
  3. Some statements in the text need to be modified, and there are some errors in the syntax.

Response:

Thanks very much for the comments, which are very helpful to improve the quality of this article. We have revised the manuscript and especially paid much attention to your comments and suggestions.

  1. We have re-checked the font format of the picture and re-uploaded it. The formula format has also been uniformly adjusted.
  2. We supplement the coupling analysis between regional airport operation capacity and regional economic and social development level. (The specific modification is located in section 5.3 on page 17)
  3.  We have corrected the grammatical mistakes in the text as much as possible.

Reviewer 3 Report

Title:

Airport group operational capacity assessment and analysis of its influencing factors: Taking Chinese typical airport group as an example.

1. What is the main question addressed by the research?

The major concern of this paper is associated with sustainability in air traffic and its management, with reference to airport capacity and its operation management. The author/ authors  try at level best to find the practical outcomes to address the topic under examine.

2. Do you consider the topic original or relevant in the field? Does it
address a specific gap in the field?

Yes, the structure of the study is relevant to the Special issue of the journal “Sustainability-MDPI”. The author/authors try to bridge the gap by using entropy methods by selecting 13 major airports of china to propose a model regarding the operation capacity and their sustainability 
3. What does it add to the subject area compared with other published
material?

This study select the busiest airport of china as study sample and apply entropy methods. The perviou studies for example “Wen lu et al” 2019 (https://www.sciencedirect.com/science/article/pii/S2092521219300781) use weighted variables and adopting CFPR. The author/authors also compare this study with https://www.hindawi.com/journals/ddns/2022/1854695/, https://www.hindawi.com/journals/jat/2022/9958810/,andhttps://mdpi res.com/d_attachment/sustainability/sustainability-12-08234/article_deploy/sustainability-12-08234-v2.pdf?version=1602573645 how this study is unique from past study and how the results bridge the gaps in past literature.

 However, in this study the defining criteria to select the variable was not defined. In this case the author use 13 variables and select them with criteria, which will address the gap in previous studies.

4. What specific improvements should the authors consider regarding the
methodology? What further controls should be considered?

The author/authors use “Tibot model”, but there is need to explain why this particular model was used, is there any specific significance to use this model in this study.

5. Are the conclusions consistent with the evidence and arguments presented
and do they address the main question posed?

Yes, the conclusions are satisfactory.

6. Are the references appropriate?

Some of the references are very old, the authors should remove and replace them with updated data, to make the study more applicable in current situation of the airports.

7. Please include any additional comments on the tables and figures.

The resolution quality of the figures still need more improvement for better graphical representation. The final version of the manuscript need a English language checking and revision/proof reading from native English.

Author Response

Reviewer 3

  1. What is the main question addressed by the research?

The major concern of this paper is associated with sustainability in air traffic and its management, with reference to airport capacity and its operation management. The author/ authors try at level best to find the practical outcomes to address the topic under examine.

  1. Do you consider the topic original or relevant in the field? Does itaddress a specific gap in the field?

Yes, the structure of the study is relevant to the Special issue of the journal “Sustainability-MDPI”. The author/authors try to bridge the gap by using entropy methods by selecting 13 major airports of china to propose a model regarding the operation capacity and their sustainability

  1. What does it add to the subject area compared with other published material?

This study select the busiest airport of china as study sample and apply entropy methods. The perviou studies for example “Wen lu et al” 2019 (https://www.sciencedirect.com/science/article/pii/S2092521219300781) use weighted variables and adopting CFPR. The author/authors also compare this study with https://www.hindawi.com/journals/ddns/2022/1854695/, https://www.hindawi.com/journals/jat/2022/9958810/,andhttps://mdpi res.com/d_attachment/sustainability/sustainability-12-08234/article_deploy/sustainability-12-08234-v2.pdf?version=1602573645 how this study is unique from past study and how the results bridge the gaps in past literature.

However, in this study the defining criteria to select the variable was not defined. In this case the author use 13 variables and select them with criteria, which will address the gap in previous studies.

  1. What specific improvements should the authors consider regarding themethodology? What further controls should be considered?

The author/authors use “Tobit model”, but there is need to explain why this particular model was used, is there any specific significance to use this model in this study.

  1. Are the conclusions consistent with the evidence and arguments presented and do they address the main question posed?

Yes, the conclusions are satisfactory.

  1. Are the references appropriate?

Some of the references are very old, the authors should remove and replace them with updated data, to make the study more applicable in current situation of the airports.

  1. Please include any additional comments on the tables and figures.

The resolution quality of the figures still need more improvement for better graphical representation. The final version of the manuscript need a English language checking and revision/proof reading from native English.

Response:

Thanks very much for the comments, which are very helpful to improve the quality of this article. We have revised the manuscript and especially paid much attention to your comments and suggestions.

1&2. Thank you for your comments on our article.

  1. Thank you for sending me links to related articles, which have benefited me a lot. About the selection of 13 variables: The research object of this paper is the main airports of China's three major airport groups, a total of 13 sample airports.
  2. We have added the background knowledge of the Tobit model, highlighted the reasons for choosing the Tobit model and added the details of the model. (The specific modification is located in section 3.2 on page 5)
  3. Thank you for your comments.
  4. We have supplemented the more recent references.
  5. We resubmitted the picture with higher resolution and corrected the grammar errors as much as possible.

Reviewer 4 Report

The paper includes interesting results, however the obtained results are of regional importance. Therefore, it can be published in any journal of China.

If authors want to publish a paper of international importance, I recommend that they obtain comparative results with other countries and regions.

Author Response

Reviewer 4

The paper includes interesting results, however the obtained results are of regional importance. Therefore, it can be published in any journal of China.

If authors want to publish a paper of international importance, I recommend that they obtain comparative results with other countries and regions.

Response:

Thank you very much for your comments. Indeed, comparisons with other countries are a valuable piece of advice. We combine previous studies on airport efficiency in other countries or regions and compare the results of this paper. Specific conclusions have been added to section 6 of the article .

Reviewer 5 Report

1. The article lacks the authors' reference to the use of the proposed method in other places in the world. No indication of similarities and differences in civil aviation traffic in China in relation to other countries.

2. Suggests presenting the research method in a graphical form showing the next steps to obtain the result.

3. I suggest presenting selected groups of airports and sub-airports on a common graphic in order to better indicate the location of these points.

4. Line 316 uses the phrase "of one airport" - please indicate which airport you are referring to.

5. In the paragraph beginning in line 57, there is a lot of interesting information about the analyzed airports. From the recipient's point of view, it would be more convenient to receive and compare these results by analyzing the data in the table. I suggest collecting and presenting this data in the form of a table.

Author Response

Reviewer 5

  1. The article lacks the authors' reference to the use of the proposed method in other places in the world. No indication of similarities and differences in civil aviation traffic in China in relation to other countries.
  2. Suggests presenting the research method in a graphical form showing the next steps to obtain the result.
  3. I suggest presenting selected groups of airports and sub-airports on a common graphic in order to better indicate the location of these points.
  4. Line 316 uses the phrase "of one airport" - please indicate which airport you are referring to.
  5. In the paragraph beginning in line 57, there is a lot of interesting information about the analyzed airports. From the recipient's point of view, it would be more convenient to receive and compare these results by analyzing the data in the table. I suggest collecting and presenting this data in the form of a table.

Response:

Thanks very much for the comments, which are very helpful to improve the quality of this article. We have revised the manuscript and especially paid much attention to your comments and suggestions.

  1. Thank you for your valuable suggestions. Supplementing the differences between civil aviation of other countries or regions and China can indeed improve this article. We combine previous studies on airport efficiency in other countries or regions and compare the results of this paper. Specific conclusions have been added to section 6 of the article.
  2. We have prepared a graphic showing the research method (Figure 1) and inserted it into page 6 of the paper.
  3. We have drawn a graph showing the location of the airport (Figure 2) and inserted it into page 7 of the article.
  4. We have modified the vague phrase "one airport" to give the specific airport name PVG.
  5. A table of airport related data information has been drawn and inserted into page 2 of the article.

Reviewer 6 Report

Dear Authors,

first of all I would like to thank you for the opportunity to read your very interesting manuscript. I have some comments/questions mentioned below:

1. Could you explain why you decided to use Tobit model instead of other regression model, I suppose that due to  variables censored in some way, could you mention which variables determined this choice. 

2. What do you mean by "ij" in formula (1)-(4), 

3. Could you explain how do you go from parameters in table 2 into entropy weight index.

Small details line 270 - put equation in the middle of line

Best regards,

Justyna Tomaszewska

Author Response

Reviewer 6

  1. Could you explain why you decided to use Tobit model instead of other regression model, I suppose that due to variables censored in some way, could you mention which variables determined this choice.
  2. What do you mean by "ij" in formula (1)-(4),
  3. Could you explain how do you go from parameters in table 2 into entropy weight index.

Small details line 270 - put equation in the middle of line

Response:

Thanks very much for the comments, which are very helpful to improve the quality of this article. We have revised the manuscript and especially paid much attention to your comments and suggestions.

  1. We have updated The background knowledge related to the Tobit model and the reason for selecting the model (The specific modification is located in section 3.2 on page 5). Of the eight variables selected, a few with insignificant results were censored.
  2. Xij is the value of the j evaluation index of the i evaluation object. We have rechecked the possible ambiguity of formula (1) - (4) and modified it.
  3. We list the indicators that define airport operation capability in Table 2, among which the data in Table 2 are descriptive statistics of each indicator. The specific value is the panel data of 13 sample airports from 2014 to 2020. Due to the large number of data, not all of them are included. According to the formula of the entropy method, the data is standardized at first, and the weight of the index is determined by calculating the information entropy of each index. Finally, the comprehensive score of the airport's operation capability is obtained.

Round 2

Reviewer 4 Report

The recent version is acceptable for publication.